# Determining the Presence of a Polymorphism in the Calpastatin (*CAST*) Gene Locus and Its Influence on Some Fattening Traits in Danube White Pigs

**DOI:** 10.3390/life16010035

**Published:** 2025-12-25

**Authors:** Katya Eneva, Radostina Stoykova-Grigorova, Gergana Yordanova, Mariyana Petrova, Radka Nedeva, Ivan Penchev, Toncho Penev

**Affiliations:** 1Agricultural Institute, 9700 Shumen, Bulgaria; katiqeneva@abv.bg (K.E.); gerganaarshspb@abv.bg (G.Y.); mari_anna1305@abv.bg (M.P.); r.nedeva@abv.bg (R.N.); 2Agricultural Academy, 1373 Sofia, Bulgaria; 3Institute of Animal Sciences-Kostinbrod, 2232 Kostinbrod, Bulgaria; rstoikova@abv.bg; 4Department of Animal husbandry—Ruminant Animals and Animal Products Technologies, Faculty of Agriculture, Trakia University, 6000 Stara Zagora, Bulgaria; ivan.penchev@trakia-uni.bg; 5Department of Ecology and Animal hygiene, Faculty of Agriculture, Trakia University, 6000 Stara Zagora, Bulgaria

**Keywords:** polymorphism, *CAST*, PCR-RFLP, *MspI*, fattening ability traits, pigs

## Abstract

A total of 57 replacement gilts of the Danube White breed was used in a study carried out in the Agricultural Institute of Shumen. During the test period, for the trait “age at reaching 90 kg live weight” as well as the following traits, they were analyzed using a Piglog 105 device (portable ultrasound scanner, Frontmatec, Denmark): back fat thickness at points X_1_ and X_2_, growth intensity, back fat of *m. Longissimus thoracis (LT)*, and lean meat percentage. DNA analysis was performed using the polymerase chain reaction restriction fragment-length polymorphism method (PCR-RFLP), with restriction endonuclease *MspI*. In the genotyped herd at the calpastatin gene locus, two alleles were identified with frequencies of 60% for allele D and 40% for allele C, and three genotypes DD, CC, and CD, with frequencies of 40%, 21%, and 39%, respectively. The percentage of animals with the DD genotype was the highest. They also had a lesser thickness of the back fat at point X_2_, a larger back fat of *LT*, and a higher percentage of lean meat.

## 1. Introduction

The calpastatin gene (*CAST*) is one of the key genes influencing skeletal muscle growth and functions as a specific inhibitor of the calpain enzyme [1]. The calpain–calpastatin proteolytic system plays an essential role in post-mortem proteolysis and therefore determines several important meat quality traits, including tenderness, water-holding capacity, and drip loss during storage [2,3].

The *CAST* acts as a major regulator of postnatal muscle growth, as it encodes calpastatin, a specific inhibitor of the calcium-dependent proteases m-calpain (macrocalpain) and µ-calpain (microcalpain) [1]. The activity of m-calpain and µ-calpain is regulated by calpastatin, which maintains the structural integrity of muscle fibers and participates in the control of cellular proliferation in vivo [4,5,6]. It has been demonstrated that increased synthesis of calpastatin, through overexpression of the *CAST*, significantly reduces calpain activity in muscle, cardiac, and neural tissues [7]. Elevated calpastatin activity can lead to the “disruption” of muscle fibers, thereby influencing meat tenderness [6,8]. Conversely, mutations in the *CAST* that reduce calpastatin expression or activity may enhance calpain activity in muscle cells, thus stimulating myoblast proliferation and the accumulation of muscle proteins, which may positively affect meat productivity.

The *CAST* has been identified as an important candidate gene for meat quality in pigs [9], poultry [10], and cattle [11,12]. In pigs, it is located on chromosome 2 (2q2.1–2q2.4) [13,14]. Several polymorphisms have been described in the *CAST*, including Arg249Lys and Ser638Arg, which are associated with meat-quality traits such as tenderness, pH, and color [14,15]. Additionally, other polymorphisms (SNPs) have been reported to affect back fat thickness in pigs [9,16,17]. Palma–Granados et al. [18] have linked the intronic variant *CAST_rs196949783 G>A* with intramuscular fat content, water-holding capacity, pH, tenderness, and *Longissimus thoracis et lumborum* thickness.

The Danube White breed was developed at the Institute of Pig Breeding through a complex synthetic breeding program involving multiple parental breeds, including Bulgarian White, Large White, Landrace, Pietrain, Hampshire, and a newly established composite group. The breed was officially recognized in 1985. Sows typically reach 250–280 kg, while boars attain 320–370 kg of live weight. Average prolificacy is 10.2 live-born piglets per litter, with an average litter weight of approximately 49 kg at 21 days of age. Animals of the Danube White breed exhibit slightly later maturity compared to modern hybrid pigs. Replacement pigs reach a live weight of 90 kg within 175–190 days and exhibit relatively thin back fat, ranging from 39 to 42 mm (CKL2 sum). The loin eye area exceeds 40 cm^2^ [19].

The present study aimed to determine the presence of polymorphism at the *CAST* locus and to evaluate its influence on traits related to fattening performance in the Danube White pigs.

## 2. Materials and Methods

### 2.1. Animals and Experimental Design

The experiment was carried out in the Agricultural Institute of Shumen with 57 replacement gilts of the Danube White breeds. The animal study protocol was approved by the Ethics Committee of Trakia University (protocol code: 428/24 March 2025).

The animals were housed at a density of 1.5 m^2^ per pig on concrete-slatted flooring, and watering was carried out with nipple drinkers installed at a height of 60 cm. A compound feed with the following ingredient composition was provided via tube feeders: corn (40%), barley (25%), wheat (10%), soybean meal (18%), sunflower meal (3%), CaCO_3_ (1%), monocalcium phosphate (0.9%), NaCl (0.3%), vitamin–mineral premix (0.8%), and energy and nutrient content (CP 15%, ME 13.0 MJ/kg, lysine 0.72%, Ca 0.80%, P 0.42%). Daily feed intake ranged between 2.3 and 2.7 kg. Compound feed/head per day depended on body condition. The microclimate was regulated by an automated electronic control system manufactured by BIG DUTCHMAN (Vechta, Lower Saxony, Germany).

Biological samples (approximately 60 hair follicles per animal) for DNA analysis were taken from the back and shoulder of the animals during the testing of 90 kg live weight and were stored in biological specimen containers (DELTALAB-PP, 150 mL, Barcelona, Catalonia, Spain), at minus 20 degrees in a freezer.

During the test period, the following traits of 90 kg live weight were analyzed in vivo using the Piglog 105 apparatus: back fat at point X_1_ (measured between the third and fourth lumbar vertebrae, 7 cm lateral; (mm)), point X_2_ (measured between the third and fourth last ribs, 7 cm lateral; (mm)), growth rate, back fat of *m. Longissimus thoracis* (measured between the third and fourth last ribs, 7 cm lateral; mm), and lean meat percentage, calculated using the multiple linear regression model [20].*LM* = 63.862 − 0.4465x_1 − 0.5096x_2 + 0.1281x_3

### 2.2. Gene Expression Analysis

Genetic analysis was carried out in the Genetics Laboratory at the Agricultural Institute of Shumen using the PCR–RFLP (polymerase chain reaction—restriction fragment length polymorphism) method. The genomic DNA of the animals included in the study was isolated using the AccuPrep Genomic DNA Extraction Kit (BIONEER Daejeon, Daejeon, Republic of Korea) according to the manufacturer’s instructions.

To amplify the studied region in the *CAST* locus, by PCR analysis, the following primer pairs were used: Forward: 5′ GCG TGC TCA TAA AGA AAA AGC 3′ and Reverse: 5′ TGC AGA TAC ACC AGT AAC AG 3′. A reaction mixture with a final volume of 22 µL was obtained, prepared in tubes with PureTaq Ready-To-Go PCR beads (96 reactions, 0.2 mL hinged tube with cap, Chicago, IL, USA).

The amplification was performed under the following conditions: initial denaturation at 94 °C for 4 min, additional denaturation at 94 °C for 1 min, amplification at 56 °C for 1 min, annealing of the fragments at 72 °C for 50 sec, repetition of the cycle (from step 2 to step 4 for 38 times), and additional amplification at 72 °C for 3 min, stored at 4 °C.

An analysis for the detection of restriction fragment length polymorphism (RFLP) was performed using restriction endonuclease *MspI*, and the samples were incubated in a thermostat (TS-100 Thermo-Shaker for microtubes and PCR plates, BioSan, Riga, Latvia) for 4 h at 37 °C.

The identification of the isolated fragments after PCR amplification and after the restriction analysis was carried out on 2.5% of agarose gel saturated with fluorescent dye RedGel (Biotium, Madrid, Spain). Obtained gels were observed on a transilluminator under UV rays using a UV transilluminator (V 2.0, WEALTEC, Miaoli, Taiwan). Numbers and lengths of the observed fragments were used to determine the corresponding genotypes of the animals. For an accurate determination of the length of the fragments in base pairs (bp), a DNA control was used—HyperLadder 100 bp (Bioline, London, United Kingdom).

### 2.3. Statistical Analyses

All statistical analyses were performed using GraphPad Prism (version 10.6.1.). Descriptive statistics were calculated for each experimental group, and data are presented as x^−^, Cv, %, R^2^. Prior to hypothesis testing, the distribution of each dataset was evaluated using the Kolmogorov–Smirnov test for normality. The differences among groups were assessed using a one-way ANOVA as factor “genotype” were applied, followed by Holm-Šídák’s multiple comparisons post hoc test. A *p*-value < 0.05 was considered statistically significant.

## 3. Results

After the polymerase chain reaction (PCR) was performed, amplified PCR products of the *CAST* were obtained in all analyzed animals.

Figure 1 presents the results of the restriction fragment length polymorphism analysis using the specific restriction endonuclease *MspI*. The alleles and corresponding genotypes were determined by the number and lengths of the fragments obtained.

Four fragments had lengths corresponding to 275, 369, 500, and 650 bp [13,19]. The C allele was characterized by three fragments of 275, 500, and 650 bp, whereas the D allele was represented by fragments of 275, 369, and 500 bp (Figure 1).

Table 1 presents allele and genotype frequencies of the studied animals of the Danube White breed. The analysis of the obtained results showed that in the genotyped herd at the locus of the *CAST* with restriction endonuclease *MspI*, two alleles were detected with frequencies, with 60% for allele D and 40% for allele C, respectively. The frequency of allele D was higher compared to that of allele C. This indicated the presence of genetic polymorphism in the studied population and a relatively even distribution between homozygous and heterozygous individuals.

In our restriction analysis, all three possible genotypes of CC, DD, and CD are observed. The lowest percentage of animals is observed in the homozygous CC genotype (21%). The other two genotypes have a similar percentage distribution, with 39% for the heterozygous CD genotype and 40% for the homozygous DD genotype.

The identified genetic variation in the *CAST* locus made it possible to determine the effectiveness of its influence on some fattening qualities of pigs in the Danube White breed herd (Table 2). Table 2 presents the results of mean values, variation, and the coefficient of determination (R^2^) for the main traits included in the assessment of self-estimation productivity in pigs from the studied herd.

For sample (*n* = 57), the obtained mean values for the back fat at the points X_1_ and X_2_ are 16.18 mm and 11.68 mm, respectively, and a low-to-medium coefficient of determination R^2^ (0.015; 0.121). The average thickness of the *Longissimus thoracis* muscle is 46.56 mm. The coefficient of determination for the studied genotype is low (R^2^ = 0.064). For the mean value of the lean meat indicator, we obtained 56.65% and a low coefficient of determination (R^2^ = 0.068).

Figure 2 presents the mean values of the main traits of the self-estimation productivity of three different genotypes of the *CAST/MspI* locus (DD, CC and CD) in pigs of the Danube White breed (*n* = 57).

The obtained data show similar phenotypic indicators between the genotypes. A statistically significant difference was found only for trait X_2_ between genotypes DD and CC (*p* < 0.05). No statistically significant differences were observed among the genotypes for the remaining traits. From the results presented in Figure 2, regarding the traits characterizing the fattening qualities in pigs, it was derived that animals homozygous for the D allele are distinguished by a smaller thickness of the back fat at point X_2_ (11.00 mm) compared to animals homozygous for the C allele (13.60 mm). At point X_1_, for the trait thickness of the back fat, a similar trend is observed, i.e., genotype CC has a higher average value (17.30 mm) compared to DD (15.82 mm) and CD (16.27 mm). The lack of significant differences suggests that the polymorphism in the *CAST* does not significantly affect the accumulation of subcutaneous fat in pigs of the Danube White breed.

Regarding the *LT* thickness trait, animals with the DD genotype (48.73 mm) were also predominant, while CC (44.40 mm) and CD (45.82 mm) exhibited lower values. The differences, although visible as a trend, were not significant, which may indicate that muscle development is more strongly influenced by other genes and factors, including nutrition and rearing conditions.

In terms of lean meat percentage, DD (57.35%) and CD (56.51%) showed higher values compared to CC (54.95%). Since the difference was not statistically significant, it cannot be assumed that there was “leaner” meat in a certain genotype of the studied population.

Figure 2 shows that heterozygous animals with the CD genotype have the oldest age at reaching 90 kg live weight (211.01 days). Animals homozygous for CC genotype have the youngest age at reaching 90 kg live weight (206.10 days).

Table 3 presents the results of a one-way analysis of variance (ANOVA) assessing the effect of genotype on the studied traits. A significant influence of genotype (*p* < 0.05) is reported only for back fat thickness at point X_2_. For the remaining traits, the genotype has no significant influence. The between-group variation (F criteria) for the reported traits is low.

## 4. Discussion

The present study confirms the presence of genetic polymorphism at the *CAST* locus in Danube White pigs, with two allele frequencies identified—D (60%) and C (40%)—and all three possible genotypes (DD, CC, and CD) (Table 1). The lowest frequency was observed for homozygous CC individuals (21%), while heterozygous CD (39%) and homozygous DD (40%) were approximately equally distributed. These results are consistent with other studies that observed similar allele and genotype frequencies in Danube White pigs and other European pig breeds [21,22,23]. Analysis of production traits indicated that *CAST* polymorphism does not exert statistically significant effects on back fat thickness (points X_1_), back fat of *LT*, and lean meat percentage. However, trends were observed, with lower back fat thickness at point X_2_, higher back fat of *LT*, and a higher percentage of lean meat in DD genotype animals (Table 2). Studies by Kluzakova et al. [21,22] assessed the genetic status of the *CAST/MspI*, *CAST/Hinf*, and *CAST/RsaI* gene loci in hybrid pig breeds. It is stated that genetic polymorphism is observed in all three loci, with the frequency of animals with the heterozygous genotype being the highest everywhere. Our results align with the findings of Škrlep et al. [24], who reported no significant differences in back fat thickness among genotypes at the *CAST* locus. In contrast, studies by Vehovský et al. [25] and Kurył et al. [26], reported the significant effect of Ser638Arg point mutation in the *CAST* on back fat thickness (*p* < 0.006) and the increased *m. Longissimus thoracis* area. According to the data in Table 2, the lean meat percentage was 56.72%, while the coefficient of determination for the genotype effect was low (R^2^ = 0.068). This value is almost identical to the lean meat percentage reported for the Bulgarian industrial population assessed using the “*S–E–U–R–O–P*” system; it obtained an average of 56.01–56.72% based on approximately 100,000 carcasses [27]. The low values of the coefficient of determination presented in Table 2 indicate that the *CAST* has only a weak influence on the quantitative traits studied. This may be due to the stronger influence of environmental factors (nutrition, rearing conditions) or a different genetic determination of these traits.

Table 3 presents the results of the analysis of variance (ANOVA). The genotype showed a significant effect only on back fat thickness at point X_2_ (*p* < 0.05), confirming the data presented in Figure 2. For the other indicators studied, no statistically significant effect of the genotype factor was reported. The data for the F criterion show that the variation between individual genotypes in the indicators studied is not large. In other words, in the studied herd of Danube White pigs, the *CAST/MspI* polymorphism did not have a statistically significant effect on the evaluated productivity traits. Although weak trends were observed between the genotypes, they were not of selection significance in the sample studied. Rybarczyk et al. [28] and Kluzakova et al. [22] found that pigs with the CD genotype showed a tendency for a higher proportion of lean meat in the carcass. Vehovský et al. [25] found a significant effect of the *Ser638Arg* point mutation in the *CAST* on lean meat content (*p* < 0.003). Other studies indicate that the *CAST* has a more pronounced effect on the qualitative and technological characteristics of meat than on growth and fattening traits [9,14].

The fattening ability of pigs from the three genotypes expressed by the trait age at reaching of 90 kg live weight was within a narrow range (206–211 days), with a tendency toward faster growth in homozygous animals—*CASTcc* (Figure 2). Stoykova–Grigorova et al. [29] established a significantly higher live weight at birth of animals homozygous for allele D (*p* < 0.05%). Despite the observed trend in Figure 2, the analysis of variance (ANOVA) presented in Table 3 shows that the genotype factor does not have a significant influence on the age at which 90 kg of live weight is reached. The data on variation among genotypes indicate that the variation in the studied indicator (age at reaching 90 kg live weight) is small.

This study presents data on the frequency of *CAST* alleles and genotypes, and examines their influence on certain productive traits in the Danube White pig breed. Although the effect on fattening traits appears limited, the presence of polymorphism and the observed trends in phenotypic characteristics justify further research. Further research should include investigations into interactions with other genes, meat quality, and the influence of environmental factors.

## 5. Conclusions

This study confirmed the presence of genetic polymorphism at the *CAST/MspI* locus in Danube White pigs and revealed the distribution of alleles and genotypes: 60% for allele D and 40% for allele C; 40% for genotype DD; 21% for genotype CC and 39% genotype CD.

The absence of statistically significant differences (except back fat thickness at point X_2_) in the sample studied limits the applicability of *CAST/MspI* as a selection marker for productive traits related to fattening performance in this population. These findings support the view that the calpastatin gene plays a more pronounced role in technological traits and post-mortem meat quality, rather than for growth and tissue accumulation in vivo.

## Figures and Tables

**Figure 1 life-16-00035-f001:**
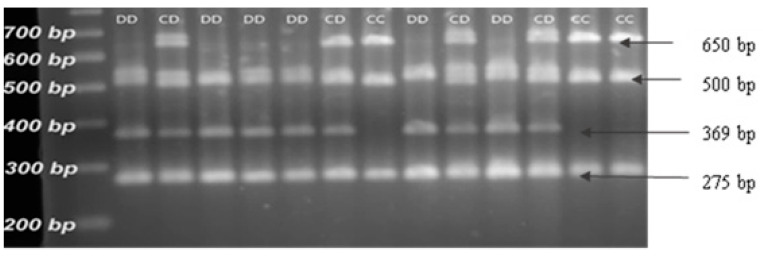
Results of the PCR-RFLP analysis of the *CAST*; visualization of the fragments on a 2.5% agarose gel after restriction of the samples with endonuclease *MspI*.

**Figure 2 life-16-00035-f002:**
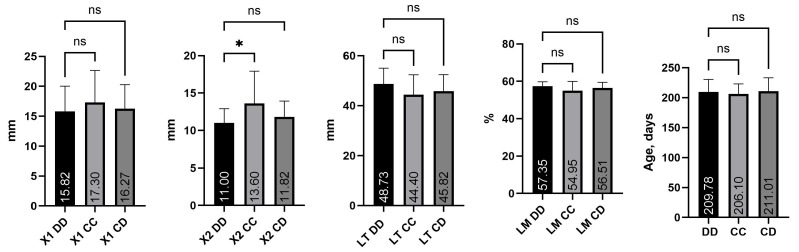
Effect of origin and genotype on productive traits. Note: ns—not significant; * significance at *p* < 0.05.

**Table 1 life-16-00035-t001:** Allelic and genotypic frequencies at the gene locus in the studied Danube White breed herd (*n* = 57).

Polymorphic Region	*CAST*/*MspI*					
Genotypes	DD		CC		CD	
Number of animals (*n*)	23		12		22	
Genotype frequencies	0.40 (40%)		0.21 (21%)		0.39 (39%)	
Allele frequencies		C = 0.40	(40%)	D = 0.60	(60%)	

*CAST*: calpastatin gene; *MspI*: restriction enzyme; genotypes: DD, CC, CD.

**Table 2 life-16-00035-t002:** Average values, variation, and coefficient of determination for traits of the self-estimation productivity (*n* = 57).

Traits	x^−^	Cv, %	R^2^
Back fat at point X_1_, mm	16.18	9.65	0.015
Back fat at point X_2_, mm	11.68	13.36	0.121
Muscle depth of *LT*, mm	46.56	13.05	0.064
Lean meat, %	56.65	1.46	0.068
Age, days	209.88	9.07	0.007

Note: Cv: coefficient of variation; R^2^—coefficient of determination.

**Table 3 life-16-00035-t003:** Analysis of variance (one-way ANOVA) for influence of factor genotype on studied parameters.

Parameters		SS	DF	MS	F	*p*-Value
X_1_	Treatment	15.10	2	7.548	F = 0.3978	*p* = 0.674
	Residual	967.7	51	18.98		
	Total	982.8	53			
X_2_	Treatment	46.48	2	23.24	F = 3.510	*p* = 0.037
	Residual	337.7	51	6.621		
	Total	384.1	53			
*LT*	Treatment	160.3	2	80.17	F = 1.761	*p* = 0.182
	Residual	2322	51	45.53		
	Total	2482	53			
*LM*	Treatment	39.71	2	19.85	F = 1.881	*p* = 0.163
	Residual	538.3	51	10.55		
	Total	578.0	53			
Age, days	Treatment	165.3	2	82.63	F = 0.1926	*p* = 0.825
	Residual	21880	51	429.0		
	Total	22045	53			

Significance at *p* < 0.05.

## Data Availability

The original contributions presented in this study are included in the article. Further inquiries can be directed to the corresponding author.

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
