# Peer review of "Determining the Presence of a Polymorphism in the Calpastatin (*CAST*) Gene Locus and Its Influence on Some Fattening Traits in Danube White Pigs"

_life, 2025, doi:10.3390/life16010035_

Round 1

Reviewer 1 Report

Comments and Suggestions for Authors
  1. Only 57 Danube White pigs were used, with uneven genotype distribution. This small sample causes all genotype-related phenotypic differences to be non-significant in Table 3,and high R² values (0.939) may be overfitting artifacts, making conclusions unconvincing.
  2. Three genotypes (DD/CC/CD) were analyzed using Student´s t-test, one-way ANOVA + post-hoc test is required for multi-group analysis.
  3. PCR-RFLP genotyping lacks key verifications (PCR product sequencing, enzyme digestion specificity).

Author Response

Response to Reviewer 1 Comments

1. Summary

2. Questions for General Evaluation

Reviewer’s Evaluation

Response and Revisions

Does the introduction provide sufficient background and include all relevant references?

Can be improved

Are all the cited references relevant to the research?

Must be improved

Is the research design appropriate?

Must be improved

Are the methods adequately described?

Must be improved

Are the results clearly presented?

Must be improved

Are the conclusions supported by the results?

Must be improved

3. Point-by-point response to Comments and Suggestions for Authors

Comments 1: Only 57 Danube White pigs were used, with uneven genotype distribution. This small sample causes all genotype-related phenotypic differences to be non-significant in Table 3,and high R² values (0.939) may be overfitting artifacts, making conclusions unconvincing.

            Three genotypes (DD/CC/CD) were analyzed using Student´s t-test, one-way ANOVA + post-hoc test is required for multi-group analysis.

            PCR-RFLP genotyping lacks key verifications (PCR product sequencing, enzyme digestion specificity).

Response 1: We agree with this comment. The Danube white breed is a meat-type breed with a limited distribution and one of the main objectives of its breeding program is to maintain genetic diversity in a population threatened with extinction.

            Following the suggested procedure, one-way ANOVA with a post-hoc test was performed, resulting in updated R2 values presented in Table 2.

Reviewer 2 Report

Comments and Suggestions for Authors

            The manuscript describes the presence of the CAST/MspI polymorphism in Danube White pigs and its potential influence on fattening traits. The topic is classical and has practical relevance, and the text is written in a clear and well-organized manner. The laboratory methods, particularly the PCR-RFLP procedure, are described correctly and allow the experiment to be reproduced. However, the manuscript requires substantial revision before it can be considered for acceptance.

            The most serious issue is the statistical analysis. The authors use Student’s t-tests to compare three genotypic groups, which is methodologically inappropriate. In studies of this type, analysis of variance (ANOVA) or a linear model with “genotype” as a factor should be applied, followed by post-hoc tests. The lack of an appropriate statistical approach means that the values presented in Table 3 cannot be regarded as fully reliable. Equally problematic are the R² values reported in Table 2. The authors provide numbers such as 0.876 and 0.939 but do not explain the exact model from which these values were derived. Given the small sample size and the absence of significant differences between genotypes, these values appear implausible and should be fully clarified or removed.

            Another important concern is the sample size. The CC homozygous group includes only 12 animals, which severely limits statistical power and in practice makes it impossible to detect subtle genetic effects. The authors should clearly acknowledge this in the discussion and treat it as a limitation of the study. Furthermore, the manuscript lacks information on important environmental and biological factors such as sex, litter structure, detailed housing conditions, or possible differences in feeding. Fattening traits are strongly influenced by the environment, and analysing them without accounting for these factors leads to an incomplete interpretation.

            The discussion is largely descriptive and summarizes previous literature instead of thoroughly interpreting the authors’ own results. The authors should address the observed lack of significance more critically, consider possible explanations, account for statistical limitations, and compare the investigated polymorphism with other, better-characterized CAST SNPs that have clearer functional relevance.

            The conclusions are correct but overly general and essentially repeat the results. It would be useful to emphasize that the absence of significant differences limits the applicability of CAST/MspI as a selection marker in this population, and that the CAST gene may play a more important role in post-mortem meat quality traits rather than in in vivo fattening indicators.

In summary, the manuscript has value as a description of the genotypic structure of a specific local population, but it requires thorough analytical and interpretative revisions. I recommend a decision of “major revision,” with the need for re-analysis of the statistical data, clarification of the methods, and strengthening of the discussion section.

Minor:

The gene name should be written in italics (e.i. line 31, 38)

l 38 – CAST has already been expanded

The first two paragraphs should be merged, as they contain overlapping information.

l 43 – m-calpain has already been explained

l 45 – in vivo should be in italics (and all Latin names later in the text: i.e. l. 71, 74)

l 69 – Remove the period after kg

l 76 – Is this a standard formula? What is its source?

l 80 – Please provide the manufacturer’s details: country and city.

l 85 – Please provide manufacturer

l 86 – Replace the semicolon with a colon.

l 95, 98 – Please provide the manufacturer’s details: country and city.

l 98 – Remove the period after ‘bp’

l 133 – add the name of the Author

Author Response

Response to Reviewer 2 Comments

1. Summary

2. Questions for General Evaluation

Reviewer’s Evaluation

Response and Revisions

Does the introduction provide sufficient background and include all relevant references?

Yes

Are all the cited references relevant to the research?

Must be improved

Is the research design appropriate?

Can be improved

Are the methods adequately described?

Can be improved

Are the results clearly presented?

Can be improved

Are the conclusions supported by the results?

Can be improved

3. Point-by-point response to Comments and Suggestions for Authors

Comments: The manuscript describes the presence of the CAST/MspI polymorphism in Danube White pigs and its potential influence on fattening traits. The topic is classical and has practical relevance, and the text is written in a clear and well-organized manner. The laboratory methods, particularly the PCR-RFLP procedure, are described correctly and allow the experiment to be reproduced. However, the manuscript requires substantial revision before it can be considered for acceptance. The most serious issue is the statistical analysis. The authors use Student’s t-tests to compare three genotypic groups, which is methodologically inappropriate. In studies of this type, analysis of variance (ANOVA) or a linear model with “genotype” as a factor should be applied, followed by post-hoc tests. The lack of an appropriate statistical approach means that the values presented in Table 3 cannot be regarded as fully reliable. Equally problematic are the R² values reported in Table 2. The authors provide numbers such as 0.876 and 0.939 but do not explain the exact model from which these values were derived. Given the small sample size and the absence of significant differences between genotypes, these values appear implausible and should be fully clarified or removed.

Response: Thank you for the recommendation. As noted by Reviewer 1, a one-way analysis of variance (ANOVA) was performed, followed by a post-hoc test. The changes in the statistical model are described in the "Statistical Analysis" section. The updated R2 values are presented in Table 2.

Comments: Another important concern is the sample size. The CC homozygous group includes only 12 animals, which severely limits statistical power and in practice makes it impossible to detect subtle genetic effects. The authors should clearly acknowledge this in the discussion and treat it as a limitation of the study. Furthermore, the manuscript lacks information on important environmental and biological factors such as sex, litter structure, detailed housing conditions, or possible differences in feeding. Fattening traits are strongly influenced by the environment, and analysing them without accounting for these factors leads to an incomplete interpretation.

Response: Thank you for your comments. Information on gender, rearing conditions and feeding (ration composition) are presented in the "Materials and Methods" section, subsection 2.1. Animals and experimental design.

Comments: The discussion is largely descriptive and summarizes previous literature instead of thoroughly interpreting the authors’ own results. The authors should address the observed lack of significance more critically, consider possible explanations, account for statistical limitations, and compare the investigated polymorphism with other, better-characterized CAST SNPs that have clearer functional relevance.

Response: We thank the reviewer for the comments. The "Results" section has been separated from the "Discussion" section. A new Figure 2 has been added in place of Table 3, presenting the variation of the studied indicators by genotype, and a new Table 3 now presents the ANOVA data. Consequently, both the "Results" and "Discussion" sections have been fully revised.

Comments: The conclusions are correct but overly general and essentially repeat the results. It would be useful to emphasize that the absence of significant differences limits the applicability of CAST/MspI as a selection marker in this population, and that the CAST gene may play a more important role in post-mortem meat quality traits rather than in in vivo fattening indicators.

Response: Thank you for your comments. The "Conclusion" section has been completely revised according to the recommendations.

The gene name should be written in italics (e.i. line 31, 38)

Response : Thank you for pointing this out. The name is written in italics

l 38 – CAST has already been expanded. The first two paragraphs should be merged, as they contain overlapping information.

Response: Thank you. These two paragraphs are merged.

l 43 – m-calpain has already been explained

Response: Thank you. The “macrocalpain” was removed

l 45 – in vivo should be in italics (and all Latin names later in the text: i.e. l. 71, 74)

Response: Thank you. The term “in vivo” was done in italics

l 69 – Remove the period after kg

Response: Thank you. The period after kg was removed

l 76 – Is this a standard formula? What is its source?

Response: Thank you. This is a standard formula, multiple linear regression model, used in pig farming to assess meat productivity and lean meat content. The author of this formula was added.

l 80 – Please provide the manufacturer’s details: country and city.

Response: Thank you. AccuPrep Genomic DNA Extraction Kit (BIONEER- Republic of Korea) was added.

l 85 – Please provide manufacturer

Response: Thank you. PureTaq Ready-To-Go PCR beads (96 reactions, 0.2 ml hinged tube with cap- UK) was added.

l 86 – Replace the semicolon with a colon.

Response: Thank you. The semicolon was replaced with a colon

l 95, 98 – Please provide the manufacturer’s details: country and city.

Response: Thank you. Hyper Ladder, 100 bp (Bioline- United Kingdom) was added.

l 98 – Remove the period after ‘bp’

Response: The period after ‘bp’ was removed.

l 133 – add the name of the Author

Response:  The name of the Author was added.  

Reviewer 3 Report

Comments and Suggestions for Authors

Dear authors,

thank you very much for submitting your manuscript. Please find the reviewer's report down below.

Kind regard.

Reviewer´s report:

Summary:

In this study the authors examine the effects of a polymorphism in the calpastatin gene on the fattening trait of Danube White pigs. They were able to show that a lower thickness of the muscle Longissimus thoracis was correlated with the genotype DD in the pigs.

General comment on the hypothesis of the work:

The present study is interesting and provides important information on the polymorphism of calpastatin gene in Danube Whit pigs.

The article addresses an important topic, but the explanations are too superficial and incomplete. It would be advisable to fundamentally revise or rewrite the article and then resubmit it for peer review.

The simple summary is missing

The abstract is too superficial. It would be better to structure the abstract as follows: Short background of the polymorphism of calpastatin and the aim of the study; Short description of the pigs and methods used; The most important results; A short discussion and conclusion based on current literature.

The introduction needs more detailed information on the breed Danube White pigs and the prevalence of polymorphisms in this breed.

In the Materials and Methods section, further additions must be made to materials and methods used. See detailed comments down below.

Results and discussion are written as a single section. Results are presented and discussed with reference to the available literature. This form of presentation is of course possible, but does not comply with the journal’s guidelines. More importantly, the way in which the results are presented with references to the literature makes them confusing and causes that important findings are lost. The presentation is confusing to read and it is difficult to separate the results from the study and those from the literature. It would be better to separate these two sections and present the details in more detail. The discussion should also refer to the literature in more detail. This will make the present work clearer and more coherent.

The reference list must be revised in accordance with the journal’s guidelines.

Comments on the Abstract:

L19: muscle Longissimus thoracis

Comments on Introduction:

LL53-57: Please describe the associations with polymorphisms in CAST gene in more detail.

Comments on Material and Methods:

LL67-68: Please list the complete diet with ingredients and nutrient components.

L69: 90 kg live weight (dot too much)

LL69-70: What did you mean with a sterile container? How did you store the samples? Please add the manufacturer, city, federal state and national state for the sterile container.

L71: Please add the manufacturer, city, federal state and national state for the Piglog 105.

L74: muscle Longissimus thoracis (italic style)

L76: Is the formula correctly represented according to mathematical rules?

L80: Please add the city, federal state and national state for the Accuprep Genomic DNA Extraction Kit.

L85: Please add the manufacturer, city, federal state and national state for the PureTaq Ready-To-Go PCR beads.

L91: Please add the manufacturer, city, federal state and national state for the thermostat.

L95: Please add the city, federal state and national state for the RedGel.

L95: Please add the manufacturer, city, federal state and national state for the transilluminator.

L100: Please add the city, federal state and national state for Microsoft Excel.

LL101-102: Please add the manufacturer, city, federal state and national state for the JASP software.

LL103-104: Please describe the significance levels.

Comments on the Results and Discussion:

LL114-117: You describe the results and add references. This paragraph is difficult to understand. Please rewrite your description.

L147: In the samples

L155: back fat

L191: [21,29]

Table 1:

Please add a description of the table and add the abbreviations CAST, MspI, DD, CC and CD and write them out.

Table 3:

Please add the calculated significance values to a new column in the table. Even if the results were not significant, the significance values are interesting for deducing possible trends.

Author Response

Response to Reviewer 3 Comments

1. Summary

2. Questions for General Evaluation

Reviewer’s Evaluation

Response and Revisions

Does the introduction provide sufficient background and include all relevant references?

Must be improved

Are all the cited references relevant to the research?

Must be improved

Is the research design appropriate?

Yes

Are the methods adequately described?

Must be improved

Are the results clearly presented?

Must be improved

Are the conclusions supported by the results?

Must be improved

3. Point-by-point response to Comments and Suggestions for Authors

Summary:

In this study the authors examine the effects of a polymorphism in the calpastatin gene on the fattening trait of Danube White pigs. They were able to show that a lower thickness of the muscle Longissimus thoracis was correlated with the genotype DD in the pigs.

General comment on the hypothesis of the work:

The present study is interesting and provides important information on the polymorphism of calpastatin gene in Danube Whit pigs.

The article addresses an important topic, but the explanations are too superficial and incomplete. It would be advisable to fundamentally revise or rewrite the article and then resubmit it for peer review.

Comments: The simple summary is missing

Response: We thank the reviewer for the constructive comment. The simple summary was written.

Comments: The abstract is too superficial. It would be better to structure the abstract as follows: Short background of the polymorphism of calpastatin and the aim of the study; Short description of the pigs and methods used; The most important results; A short discussion and conclusion based on current literature.

Response: We thank the reviewer for the constructive comment. The Abstract was fully revised.

Comments: The introduction needs more detailed information on the breed Danube White pigs and the prevalence of polymorphisms in this breed.

Response: The information on the breed Danube White pigs was written. The prevalence about polymorphism in this breed was added in “Discussion” section.

Comments: In the Materials and Methods section, further additions must be made to materials and methods used. See detailed comments down below.

Response: Section “Material and methods” was fully revised according to reviewer 1 and 2 suggestions.

Comments: Results and discussion are written as a single section. Results are presented and discussed with reference to the available literature. This form of presentation is of course possible, but does not comply with the journal’s guidelines. More importantly, the way in which the results are presented with references to the literature makes them confusing and causes that important findings are lost. The presentation is confusing to read and it is difficult to separate the results from the study and those from the literature. It would be better to separate these two sections and present the details in more detail. The discussion should also refer to the literature in more detail. This will make the present work clearer and more coherent.

The reference list must be revised in accordance with the journal’s guidelines.

Response: We thank the reviewer for the constructive comment. The section Results was separated from section Discussion. After new statistical analysis some values are changed and this is pointed with tables and figures. Section Discussion is fully rewritten. All changes are in red color.

Comments on the Abstract:

L19: muscle Longissimus thoracis

Response: Thank you for pointing this out. This suggestion are made. m. Longissimus thoracis ()

Comments on Introduction:

LL53-57: Please describe the associations with polymorphisms in CAST gene in more detail.

Response: Thank you for the note. “Studies by other authors have linked the presence of the intronic variant CAST_rs196949783G>A to intramuscular fat content, water retention capacity of the meat, pH, tenderness and thickness of Longissimus thoracis et lumborum (Palma-Granados et, al., 2024)” was added.

Comments on Material and Methods:

LL67-68: Please list the complete diet with ingredients and nutrient components.

Response: Thank you. “Compound feed with ingredient composition (corn-40%, barley-25%, wheat-10%, soybean meal-18%, sunflower meal-3%, CaCo3-1%, monocalcium phosphate-0.9%, NaCl-0.3%, VMP-0.8%) and energy and nutrient content (SP-15%, CE-13.0 MJ/kg, lysine-0.72%, Ca-0.80%, P-0.42%) was consumed from tube feeders” was added.

Comments: L69: 90 kg live weight (dot too much)

Response: The Danube White breed is a meat breed, which is characterized by excellent adaptive and meat qualities and late maturity. For this reason, the testing of animals is carried out at 90 kg live weight.

Comments: LL69-70: What did you mean with a sterile container? How did you store the samples? Please add the manufacturer, city, federal state and national state for the sterile container.

Response: “Biological specimen container for storing biological samples. The hair follicles were stored in sterile containers at minus 20 degrees in a freezer. Biological specimen container (DELTALAB-PP, 150 ml- Spain)” was added.

Comments: L71: Please add the manufacturer, city, federal state and national state for the Piglog 105.

Response: Thank you for pointing this out. “Piglog 105 (portable ultrasound scanner - Frontmatec (Denmark)” was added.

Comments: L74: muscle Longissimus thoracis (italic style)

Response: Thank you for pointing this out “m. Longissimus thoracis ()” was added.

Comments: L76: Is the formula correctly represented according to mathematical rules?

Response: Thank you for this note: This is a standard formula, multiple linear regression model. The Authors of this formula was added.

Comments: L80: Please add the city, federal state and national state for the Accuprep Genomic DNA Extraction Kit.

Response: Thank you. “AccuPrep Genomic DNA Extraction Kit (BIONEER- Republic of Korea)” was added.

Comments: L85: Please add the manufacturer, city, federal state and national state for the PureTaq Ready-To-Go PCR beads.

Response: Thank you. “PureTaq Ready-To-Go PCR beads (96 reactions, 0.2 ml hinged tube with cap- UK)” was added.  

Comments: L91: Please add the manufacturer, city, federal state and national state for the thermostat.

Response: Thank you. “The thermostat (TS-100, Thermo-Shaker for microtubes and PCR plates-BioSan-Latvia) for 4 hours at 37 °C” was added.

Comments: L95: Please add the city, federal state and national state for the RedGel.

Response: Thank you. “RedGel (Biotium-U.S)” was added.

Comments: L95: Please add the manufacturer, city, federal state and national state for the transilluminator.

Response: Thank you. The “UV transilluminator V 2.0- WEALTEC- Taiwan” was added.

Comments: L100: Please add the city, federal state and national state for Microsoft Excel.

Response: Thank you. This paragraph was fully changed.

Comments: LL101-102: Please add the manufacturer, city, federal state and national state for the JASP software.

Response: Thank you. The statistical program was changed.

Comments: LL103-104: Please describe the significance levels.

Response: It was written new table 3 of analysis of variance ANOVA. All significance levels are described.

Comments on the Results and Discussion:

LL114-117: You describe the results and add references. This paragraph is difficult to understand. Please rewrite your description.

Response: Thank you. The paragraph Results and paragraph Discussion was fully rewritten.

Comments: L147: In the samples

Response: Thank you. The “sample (n = 57)” was added.

Comments: L155: back fat

Response: Thank you. “Backfat thickness” was added

Comments: L191: [21,29]

Response: Thank you. It is changed according journal requirements. 

Comments: Table 1:

Please add a description of the table and add the abbreviations CAST, MspI, DD, CC and CD and write them out.

Response: Thank you. The sentence “CAST-calpastatin gene, MspI-restriction endonuclease, genotypes; DD, CC and CD” was added.

Comments: Table 3: Please add the calculated significance values to a new column in the table. Even if the results were not significant, the significance values are interesting for deducing possible trends.

Response: Thank you. The new table was added (Table 3)

Round 2

Reviewer 1 Report

Comments and Suggestions for Authors Include representative images of the studied pigs.                

Author Response

Reviewer sugestions are made,

The picture of studied pigs is submited.

Reviewer 2 Report

Comments and Suggestions for Authors

The authors have addressed my comments, and I have no further remarks.

Author Response

Thank you to reviewer comments. 

Reviewer 3 Report

Comments and Suggestions for Authors

Dear authors,

thank you very much for incorporating my comments. The additions to the abstract and introduction make the manuscript easier to understand. The description of the breeding of the Danube pig is a particularly important aspect. Dividing the results and discussion into separate sections clearly highlights the results and gives the manuscript a clear structure. I have only a few further comments.

Kind regards.

Comments:

L64: better: Palma-Granados et al. [18] have linked…

L67: thickness (not in italic style)

L82: Please add the ethical statement in LL309-310 to the description of the experiment in the Material and Methods section too.

L90: Did you mean 2.3-2.7 kg/sow/day?

L92: Please add city, federal and national state for Big Dutchman.

L95: Please add city and federal state of the manufacturer of Deltalab-PP.

LL108-109: Please add city and federal state of the manufacturer of AccuPrep Genomic DNA Extraction Kit.

LL113-114: Please add city and federal state of the manufacturer of PureTaq Ready-To-Go PCR beads.

L121: Please add city and federal state of the manufacturer of TS-100 Thermo-Shaker.

L124: Please add city and federal state of the manufacturer of RedGel.

L125: Please add city and federal state of the manufacturer of UV Transilluminator.

L128: Please add city and federal state of the manufacturer of HyperLadder 100 bp.

L165: (Table 2) not (Tables 2)

L173: the Longissimus thoracis (the not in italic style)

L232-233: In studies by Kluzakova et al. [21,22]…

L240: Longissimus (capital letter, italic style) Please specify the muscle. Did you mean the muscle Longissimus thoracis?

L243: Please write out the abbreviation SEUROP.

L257: Better: Rybarczyk et al. [28] and Kluzakova et al. [22]… (delete “Some authors”)

L265: Figure 2 (capital letter)

Table 3:

Please check the formatting of the table.

Please use the decimal point instead of comma for the values in the table.

Author Response

Comments:

L64: better: Palma-Granados et al. [18] have linked…

L67: thickness (not in italic style)

L82: Please add the ethical statement in LL309-310 to the description of the experiment in the Material and Methods section too.

L90: Did you mean 2.3-2.7 kg/sow/day?

L92: Please add city, federal and national state for Big Dutchman.

L95: Please add city and federal state of the manufacturer of Deltalab-PP.

LL108-109: Please add city and federal state of the manufacturer of AccuPrep Genomic DNA Extraction Kit.

LL113-114: Please add city and federal state of the manufacturer of PureTaq Ready-To-Go PCR beads.

L121: Please add city and federal state of the manufacturer of TS-100 Thermo-Shaker.

L124: Please add city and federal state of the manufacturer of RedGel.

L125: Please add city and federal state of the manufacturer of UV Transilluminator.

L128: Please add city and federal state of the manufacturer of HyperLadder 100 bp.

L165: (Table 2) not (Tables 2)

L173: the Longissimus thoracis (the not in italic style)

L232-233: In studies by Kluzakova et al. [21,22]…

L240: Longissimus (capital letter, italic style) Please specify the muscle. Did you mean the muscle Longissimus thoracis?

L243: Please write out the abbreviation SEUROP.

L257: Better: Rybarczyk et al. [28] and Kluzakova et al. [22]… (delete “Some authors”)

L265: Figure 2 (capital letter)

Table 3:

Please check the formatting of the table.

Please use the decimal point instead of comma for the values in the table.

We would like to thank you for your review and for your support in facilitating the publication of our manuscript entitled "Determining the presence of a polymorphism in the calpastatin (CAST) gene locus and its influence on some fattening traits in Danube White pigs." The revisions we have made fully address your comments (yellow background).